# Functional evaluation of transposable elements as enhancers in mouse embryonic and trophoblast stem cells

Christopher D Todd[1,2], Özgen Deniz[1,2], Darren Taylor[2], Miguel R Branco[2]*

[1]Blizard Institute, Barts and The London School of Medicine and Dentistry, Queen Mary University of London, London, United Kingdom; [2]Centre for Genomic Health, Life Sciences Institute, Queen Mary University of London, London, United Kingdom

**Abstract** Transposable elements (TEs) are thought to have helped establish gene regulatory networks. Both the embryonic and extraembryonic lineages of the early mouse embryo have seemingly co-opted TEs as enhancers, but there is little evidence that they play significant roles in gene regulation. Here we tested a set of long terminal repeat TE families for roles as enhancers in mouse embryonic and trophoblast stem cells. Epigenomic and transcriptomic data suggested that a large number of TEs helped to establish tissue-specific gene expression programmes. Genetic editing of individual TEs confirmed a subset of these regulatory relationships. However, a wider survey via CRISPR interference of RLTR13D6 elements in embryonic stem cells revealed that only a minority play significant roles in gene regulation. Our results suggest that a subset of TEs are important for gene regulation in early mouse development, and highlight the importance of functional experiments when evaluating gene regulatory roles of TEs.

*For correspondence:
m.branco@qmul.ac.uk

Competing interests: The authors declare that no competing interests exist.

## Introduction

Our knowledge of the tissue-specific regulatory landscape of genomes has vastly increased over the last 10 years, thanks in part to large efforts from consortia such as ENCODE and NIH Roadmap (*ENCODE Project Consortium, 2012*; *Kundaje et al., 2015*). But whilst such mapping efforts have been instrumental in categorising the non-coding genome into different types of biochemical activity, our understanding of the associated functional roles remains limited. One of the grand challenges of the post-ENCODE era has been to ascribe regulatory function to the biochemically active non-coding portion of genomes.

This question is particularly pertinent to transposable elements (TEs) (*Elliott et al., 2014*; *Doolittle and Brunet, 2017*), which often display marks of regulatory activity in a species-specific manner (*Jacques et al., 2013*; *Carninci, 2014*). On one hand, successful TEs are expected to display such active profiles, which serve the selfish interests of TEs but may act neutrally with respect to host fitness. On the other, TEs can be co-opted (or exapted) by the host to serve gene regulatory roles, such as alternative promoters or enhancers (*Chuong et al., 2017*). The distinction between these two scenarios relies on approaches that query the causal links between TEs, gene expression and phenotype.

Primarily based on epigenomic data, evidence to date suggests that particular TE families have contributed to the evolution of tissue-specific gene regulatory networks in contexts such as early development (*Kunarso et al., 2010*), placentation (*Chuong et al., 2013*), pregnancy (*Lynch et al., 2011*) and innate immunity (*Chuong et al., 2016*), amongst others. Transcription factor (TF) binding motifs hosted within the regulatory portion of TEs enable their expression in specific tissues (*Sundaram et al., 2014*; *Sundaram et al., 2017*), presumably in a manner that enables vertical inheritance of new TE insertions via the germline. In this respect, mammalian preimplantation

**eLife digest** Much of what is known about genetics has come from studying only a tiny fraction of the genome's sequence, the part that primarily codes for proteins. But the genome has many other features outside these regions, some of which play an important biological role. Transposable elements – repetitive sequences that are present in many species – make up around half of the mouse genome. They are 'selfish' elements, in that the spread of them within the genome does not necessarily benefit the host organism. But sometimes transposable elements can be 'domesticated', and used to the host's advantage. For example, transposable elements can generate new genes. In other cases, their non-coding sequences can regulate the activity of other nearby genes or even those elsewhere in the genome.

It remains unclear to what extent transposable elements have shaped genome regulation throughout evolution. One idea is that the spread of transposable elements can help to establish large regulatory networks – whereby many genes are collectively regulated to produce a specific output. But it has not been fully explored how effective transposable elements are at regulating gene expression. Now, Todd et al. investigate whether particular transposable elements, that are suspected to boost the activity of other genes, are essential for normal gene expression in early mouse development.

Todd et al. genetically edited stem cells from the inner and outer layer of the early mouse embryo to find transposable elements that promote gene expression. Whilst some transposable elements were found to be important for gene regulation, not all of the candidates tested were needed to maintain expression levels. To widen the search, several transposable elements were turned off simultaneously by compacting specific stretches of DNA so that they could no longer be activated. When 34 transposable elements were inactivated at once, it emerged that only three transposable elements had a significant impact on gene expression. These findings suggest that whether or not a given transposable element regulates gene expression cannot be predicted solely from profiling the structure and sequence of the genome. This highlights why it is important to interrogate the effect transposable elements have on a gene's role within a cell.

Transposable elements are largely disregarded in genomics due to technical difficulties in analysing these repetitive stretches of DNA. But characteristic variations within a population may in part be driven by differences in these parts of the genome, which may also be implicated in diseases such as cancer. Identifying which transposable elements are important for driving gene expression, and linking their actions to specific traits could aid the discovery of important genetic variants.

development is a seemingly well exploited context for TE expansion, driving genetic conflicts with the host, as well as creating opportunities for TE exaptation (*Rodriguez-Terrones and Torres-Padilla, 2018*). In the mouse, TE-derived regulatory activity has been implicated at multiple stages of preimplantation development. Namely, MERVL elements become highly activated upon zygotic genome activation and are thought to play a role in the establishment of the 2-cell stage expression programme (*Macfarlan et al., 2012*). Transition from the 2-cell stage and development progression to the blastocyst stage appear to depend on LINE-1 expression (*Jachowicz et al., 2017*; *Percharde et al., 2018*). Finally, work from embryonic and trophoblast stem cells (ESCs and TSCs, respectively), suggests a divergence in TE regulatory activity that is concomitant with the separation of the embryonic and extraembryonic lineages at the blastocyst stage (*Kunarso et al., 2010*; *Chuong et al., 2013*). In ESCs, TE families such as RLTR13D6 bind key ESC TFs (e.g., OCT4, NANOG), whereas a distinct subset (e.g., RLTR13D5) bind factors essential for the maintenance of the TSC state (CDX2, ELF5, EOMES). These elements are enriched for histone marks that are characteristic of distal enhancers and lie near genes that are expressed in the lineages where they are active (*Kunarso et al., 2010*; *Chuong et al., 2013*). TE enhancer activity depends on the cooperative action of multiple TFs, whose binding motifs appear to have been already present in the corresponding ancestral TE insertions (*Sundaram et al., 2017*). However, it remains unclear to what extent such lineage-specific TEs are important for maintaining gene expression programmes during preimplantation development.

Here we have tested the gene regulatory function of specific TE families in ESCs and TSCs using genetic and epigenetic editing approaches, and compared them with predictions from extensive analyses of epigenomic and transcriptomic data. We identify a number of TEs that are important to drive expression of lineage-specific genes. However, our data suggest that these constitute a minority of all the putative TE-derived enhancers identified through bioinformatic analyses, highlighting the importance of functional tests when assessing the contribution of TEs to gene regulatory networks.

## Results

### TE-derived enhancers in ESCs and TSCs are highly tissue-specific

To identify TEs with putative regulatory potential in embryonic and extraembryonic lineages of the blastocyst (*Figure 1A*), we focused on a set of TE families that were previously shown to be highly enriched for binding of key TFs in ESCs (RLTR9, RLTR13D6) (*Kunarso et al., 2010*) or TSCs (RLTR13B, RLTR13D5) (*Chuong et al., 2013*). These long terminal repeat (LTR) families entered the Muridae lineage within the last ~12.5 million years (*Thybert et al., 2018*), with copy numbers for each subfamily ranging from 35 (RLTR9A4) to 1302 (RLTR9E), according to the Repeatmasker annotation (*Figure 1—figure supplement 1A,B*). The majority of these elements are found as solo LTRs, as judged by the genomic distance, length and arrangement of consecutive LTRs (*Figure 1—figure supplement 1C*). Nonetheless, we identified some putative proviral elements, mainly associated with RLTR9 subfamilies (*Figure 1—figure supplement 1C*). These include MMERVK9C elements (bearing RLTR9C LTRs) and MMERVK9E elements (RLTR9E LTRs). For this study we considered all LTR copies, irrespective of their genomic arrangement.

Using uniquely aligned reads from publicly available sequencing data (*Supplementary file 1*), we selected TEs bearing the hallmarks of enhancer elements, namely open chromatin status (from ATAC-seq data), binding of at least one of three key TFs (NANOG, OCT4 or SOX2 in ESCs; ELF5, EOMES or CDX2 in TSCs) and enrichment for H3K27ac. To stringently rule out gene promoters we excluded TEs enriched for H3K4me3 and/or lying within 500 bp of known mRNA transcriptional start sites. These putative 'TE+ enhancers' also displayed H3K4me1 marking (*Figure 1B*) and were bound by multiple proteins normally associated with enhancer activity, such as p300 and the Mediator and cohesin complexes (*Figure 1—figure supplement 2A*). This stringent selection led to the identification of 634 TE+ enhancers in ESCs and 358 in TSCs, which represent respectively 9.6% and 13% of all the TE copies in the families considered. To estimate how many TE+ enhancers are potentially missed due to reduced read mappability, we simulated sequencing reads that produced ChIP-seq peaks at every TE of interest. After remapping, peak detection failed for 597 RLTR13D6/RLTR9 elements and 915 RLTR13D5/RLTR13B elements, raising the possibility that an additional substantial fraction of TEs from these families could harbour enhancer marks.

As expected, RLTR13D6/RLTR9 elements only displayed enhancer-like profiles in ESCs and not in TSCs, whereas the reverse was true for RLTR13D5/RLTR13B elements (*Figure 1C*). To confirm that TE+ enhancers display a similar lineage asymmetry in vivo and assess the timing of enhancer activation, we analysed two ATAC-seq datasets from pre- and post-implantation embryos (*Wu et al., 2016*; *Smith et al., 2017*). We found that 49% and 56% of the identified TE+ enhancers (in TSCs and ESCs, respectively) display open chromatin in vivo at some point in early development, with a subset being already active in preimplantation embryos (*Figure 1D*). Both ESC and TSC TE + enhancers show some asymmetry with respect to their chromatin status in the respective post-implantation lineages (epiblast and extraembryonic ectoderm), but the tissue specificity is more pronounced for TSC TE+ enhancers (*Figure 1D*).

Following from the above observations and previous findings (*Jacques et al., 2013*), we asked more widely whether TE+ enhancers displayed tissue-specific open chromatin. For comparison, we also generated a list of 'TE- enhancers' with the same characteristics as TE+ enhancers (*Figure 1—figure supplement 2B–D*) but that did not overlap any repetitive elements within the Repeatmasker annotation (yielding 1988 elements in ESCs and 319 in TSCs). Strikingly, whilst a substantial proportion of ESC TE- enhancers displayed open chromatin in multiple tissues, TE+ enhancer activity was far more restricted to ESCs (*Figure 1E*). Similar results were obtained for TSC enhancers (*Figure 1—figure supplement 3*), in line with previous work (*Chuong et al., 2013*). These results suggest that

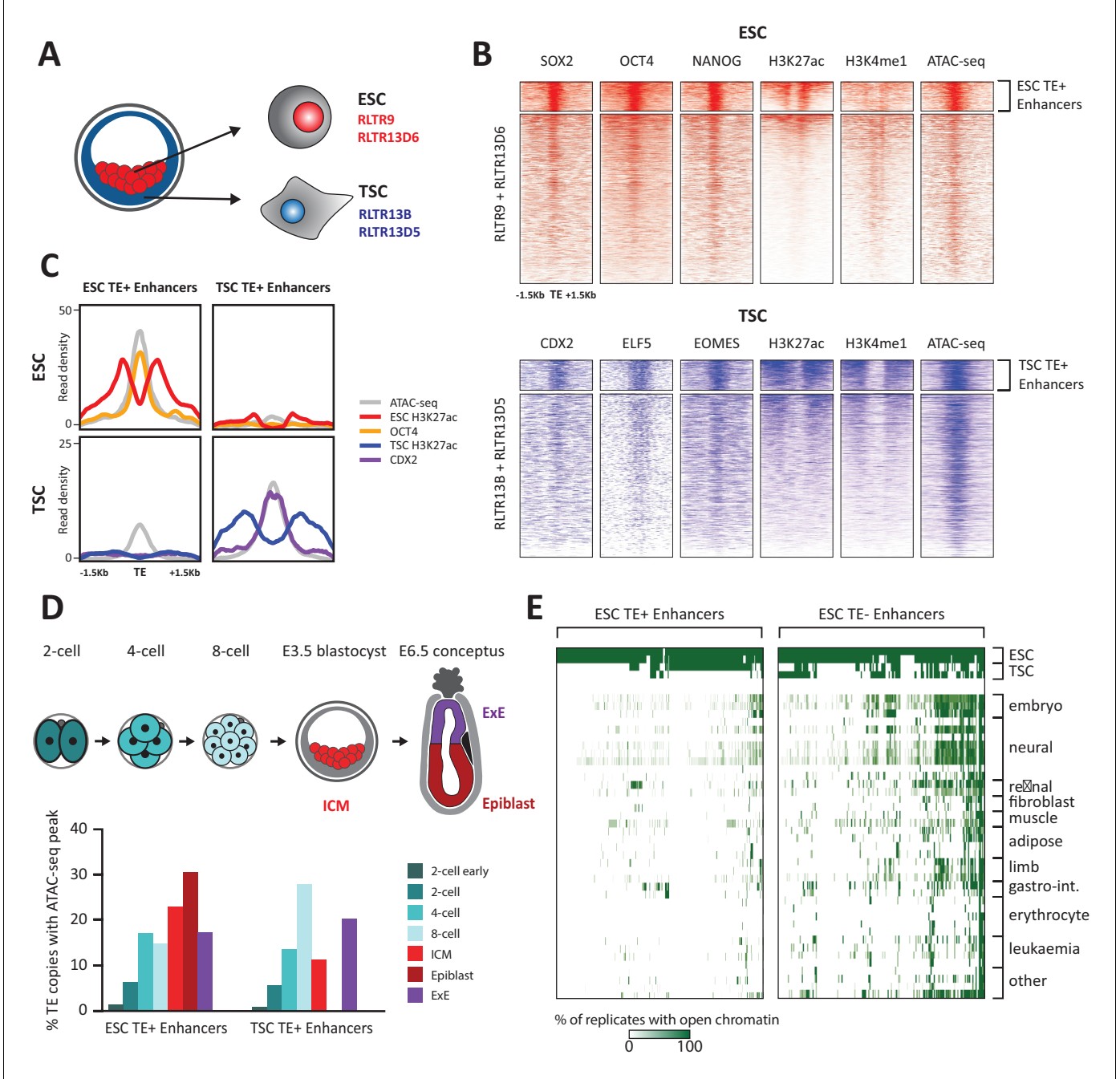

**Figure 1.** TE+ enhancers are highly tissue-specific. (**A**) We focused on specific TE families that were previously associated with enhancer activity in ESCs and TSCs. (**B**) ChIP-seq and ATAC-seq profiles show how the selected ESC and TSC TE+ enhancers are highly enriched for cell-specific TFs, enhancer-associated chromatin marks, and open chromatin. (**C**) Comparison of average TE+ enhancer ChIP-seq and ATAC-seq profiles between ESCs and TSCs demonstrate that enhancer-like profiles are cell-specific with respect to each group of enhancer families. (**D**) Analysis of ATAC-seq data from pre- and post-implantation embryos, displaying the percentage of ESC and TSC TE+ enhancers with open chromatin in each stage/tissue. (**E**) Analysis of ENCODE DNase-seq data from multiple tissues, displaying overlaps of open chromatin regions with TE+ and TE- enhancers in ESCs. For each TE, the colour intensity is proportional to the percentage of replicates of the same tissue that overlapped an open chromatin region.

The online version of this article includes the following figure supplement(s) for figure 1:

**Figure supplement 1.** Genomic characteristics of selected TE families.

**Figure supplement 2.** Additional chromatin profiles of TEs and enhancers.

**Figure supplement 3.** Tissue specificity of TSC TE+ enhancers.

TE+ enhancers are particularly optimised for activity within their respective early embryonic tissues, possibly through the synergistic action of multiple TF binding events (*Sundaram et al., 2017*). Co-option of TE+ enhancers may therefore particularly benefit genes that require highly tissue-specific expression.

## TE enhancer activity cannot be faithfully predicted from TF binding motifs

Despite their sequence similarity, only a relatively small fraction of TEs from any given family bear enhancer-like profiles. It was previously suggested that TE enhancer activity in the ESC and TSC contexts is determined by the presence of key TF binding motifs, which have otherwise been mutated in inactive TEs (*Chuong et al., 2013*; *Sundaram et al., 2017*). However, it remains unclear whether such motifs are fully determinant of TE enhancer activity. We therefore performed TF motif analyses of TE+ enhancer sequences. For comparison, we identified TEs from the same families with high read mappability but that did not display enhancer marks, henceforth termed 'non-enhancer TEs' (*Figure 2—figure supplement 1A*). TE+ enhancers are on average longer than non-enhancer TEs, as expected if deletions remove key TF binding sites (*Figure 2A,B*; *Figure 2—figure supplement 1B*). Nevertheless, there are many non-enhancer TEs that are full-length LTRs, prompting the question of what are the sequence determinants of enhancer activity. Focusing on long LTR elements (>60% of maximum length for each family), we found enrichment of several motifs at TE+ enhancers (versus non-enhancer TEs). However, no single motif was predictive of enhancer-like profiles (*Figure 2A,B*; *Figure 2—figure supplement 1B*). For example, whilst SOX2 binding motifs were present in nearly all (81–91%) RLTR13D6 and RLTR9E TE+ enhancers, a high proportion (48–58%) of non-enhancer TEs also contained this motif. Notably, motifs for other TFs (ESRRB, KLF4) that have been shown to cooperate with SOX2 for RLTR9E enhancer activity (*Sundaram et al., 2017*) were present in similar abundance at both enhancer and non-enhancer TEs (*Figure 2A*). The co-occurrence of multiple TF motifs in the same element was also insufficient to fully predict enhancer-like profiles. For example, elements containing OCT4, SOX2 and NANOG motifs accounted for 65% of TE+ enhancers and 21% of non-enhancer TEs in the RLTR13D6 family (*Figure 2B*), and few RLTR13D5 elements contained all three EOMES, ELF5 and CDX2 motifs (*Figure 2—figure supplement 1B*).

We then asked whether TF binding motifs predicted plasmid-based enhancer activity better than they predict enhancer-like chromatin profiles. Using data from a high-throughput reporter assay in ESCs (*Murtha et al., 2014*), we found that SOX2, OCT4 and NANOG binding motifs were present in only 12–30% of TEs with plasmid-based enhancer activity (*Figure 2C*). This suggests that simple sequence features, such as the motifs considered here, are poor predictors of TE enhancer activity, which is in line with recent findings in human ESC enhancers (*Barakat et al., 2018*).

Notably, 64% of RLTR13D6/RLTR9 copies with enhancer activity in the reporter assay did not display an enhancer-like chromatin profile. We therefore asked whether TF binding and chromatin opening at non-enhancer TEs was repressed by chromatin features. As we previously described (*de la Rica et al., 2016*), non-enhancer TEs display higher levels of DNA methylation than TE+ enhancers (*Figure 2D*). However, removal of DNA methylation did not lead to increased enhancer activity at non-enhancer TEs, as judged from ATAC-seq and ChIP-seq data from ESCs lacking DNA methylation (triple knockout of *Dnmt1/3a/3b*; *Figure 2E*) (*Domcke et al., 2015*). Similar results were obtained with data from hypomethylated naïve ESCs (*Figure 2—figure supplement 1C*) (*Kim et al., 2018*). Additionally, we found no evidence of other chromatin marks that could be maintaining TE enhancer activity repressed (*Figure 2—figure supplement 1D*).

All together, these data show that TE enhancer capacity appears to behave non-deterministically with respect to TF motifs or repressive chromatin marks. Therefore, whilst chromatin profiling and reporter assays are useful probabilistic indicators of potential enhancer activity, the regulatory action of TEs has to ultimately be tested through molecular manipulations in their genomic environment.

## TE-derived enhancers interact with lineage-specific genes

To establish correlations between TE+ enhancer activity and gene expression, studies to date have largely relied on the linear proximity between TEs and genes. This disregards 3D genome conformation, which enables long-range interactions and is not restricted to one-to-one relationships between TEs and genes. We therefore coupled TE+ enhancers to genes they putatively regulate based on

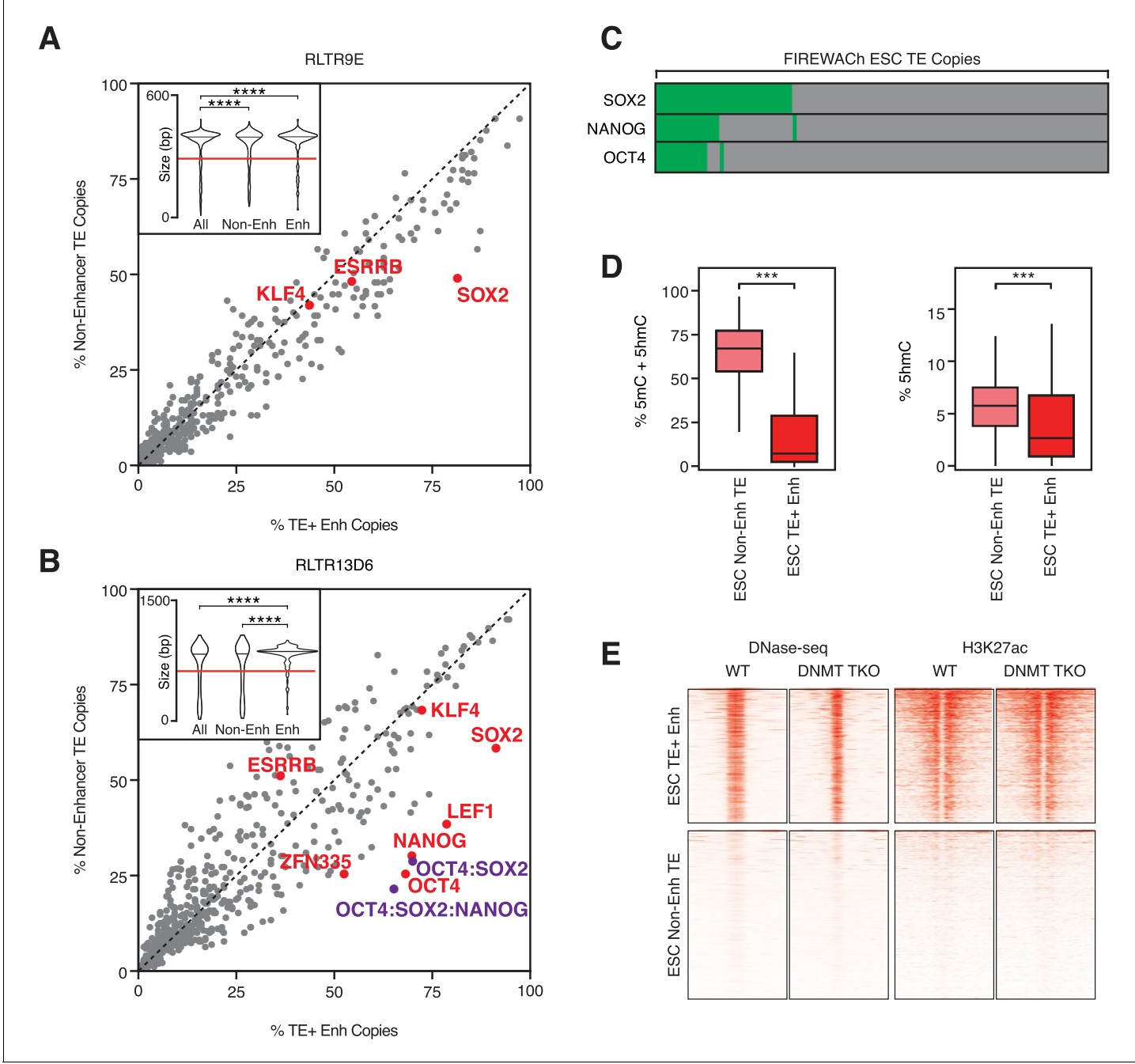

**Figure 2.** TF motifs do not predict TE enhancer potential. (**A,B**) Abundance of TF motifs at TE+ enhancers and non-enhancer TEs from RLTR9E (**A**) and RLTR13D6 (**B**) families. Purple data points refer to the co-occurrence of multiple TF motifs in the same element. Inset: length distribution of elements in each category (****p<0.0001, Kolmogorov-Smirnov test); only elements longer than 60% of the maximum length (red line) were used for motif analysis. (**C**) A minority of TE copies with enhancer activity in a reporter-based assay (FIREWACh) harbour SOX2, NANOG or OCT4 motifs, as indicated in green. (**D**) Analysis of BS-seq and TAB-seq data shows that non-enhancer TEs display higher levels of DNA methylation than TE+ enhancers and only moderately higher levels of hydroxymethylation (***p<1E-10, Wilcoxon test). (**E**) DNase-seq and H3K27ac ChIP-seq profiles of TE+ enhancers and non-enhancer TEs in wildtype and *Dnmt* TKO ESCs, showing similar profiles between both cell lines.

The online version of this article includes the following figure supplement(s) for figure 2:

**Figure supplement 1.** Additional comparisons between TE+ enhancers and non-enhancer TEs.

promoter capture Hi-C (PCHi-C) data that we recently generated in ESCs and TSCs (*Schoenfelder et al., 2018*). Only 34–44% of TE+ enhancers interacted with at least one gene promoter, which was nonetheless higher than the proportion of non-enhancer TEs with gene promoter interactions (21–28%, *Figure 3—figure supplement 1A*). In contrast, a high proportion of TE-enhancers (65–70%) interacted with gene promoters. The contrast between TE+ and TE- enhancers could be explained by the fact that the latter are preferentially positioned within gene-rich, active regions (known as the 'A' spatial compartment), whereas TE+ enhancers tend to be located within gene-poor, inactive regions ('B' compartment; *Figure 3—figure supplement 1B*). Accordingly, TE + enhancers and TE- enhancers interact with largely non-overlapping groups of genes (*Figure 3—figure supplement 1C*).

To analyse correlations between enhancers and gene expression, we only considered genes that interact exclusively with TE+ or TE- enhancers (*Figure 3A*). For both ESCs and TSCs, we found that TE+ enhancers interacted with genes that displayed higher expression levels when compared to the genome-wide average or to genes interacting with non-enhancer TEs (*Figure 3B*). Given the tight

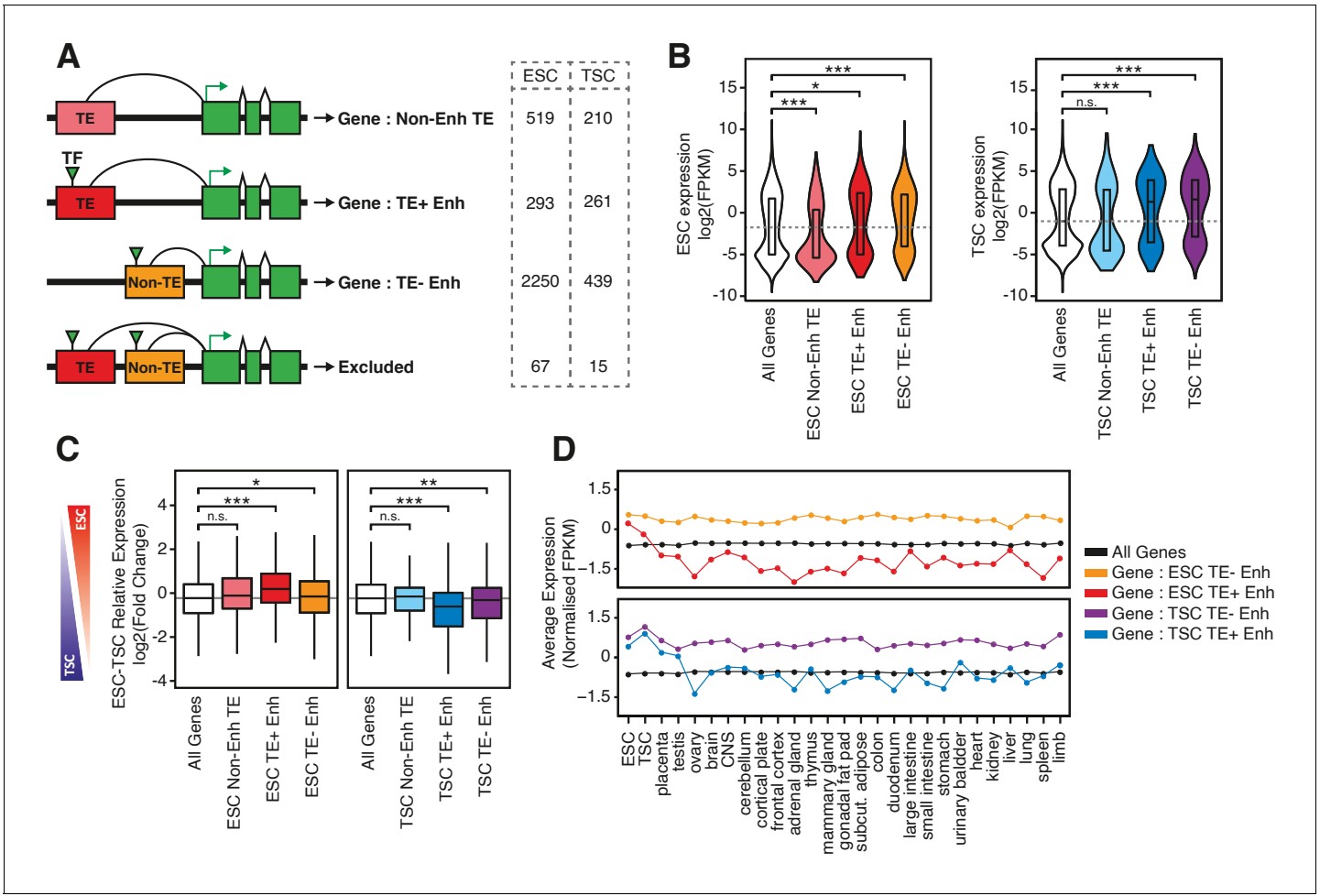

**Figure 3.** TE+ enhancers interact with tissue-specific genes. (A) Different types of elements were linked with genes and their expression levels based on PCHi-C data. Genes interacting with both TE+ and TE- enhancers were not considered. Numbers of gene promoters in each group are shown. (B) ESC and TSC expression levels for genes interacting with each of the groups indicated in A, and compared with the full distribution of expression levels (*p<0.05, ***p<0.001, ANOVA with Holm-Sidak's post-hoc test). (C) Relative expression levels (ESC/TSC ratio) for genes expressed in at least one of the two cell types (*p<0.05, **p<0.01, ***p<0.001, ANOVA with Holm-Sidak's test). (D) Average expression levels of ENCODE RNA-seq datasets from multiple tissues, compared with data from ESCs and TSCs. Data were normalised through distribution matching.

The online version of this article includes the following figure supplement(s) for figure 3:

**Figure supplement 1.** Spatial arrangement of TEs and enhancers.

**Figure supplement 2.** Gene expression changes during early differentiation.

tissue specificity of TE+ enhancer profiles that we described above (*Figure 1E*; *Figure 1—figure supplement 3*), we asked whether genes interacting with TE+ enhancers displayed lineage-specific expression. We first compared their expression in ESCs and TSCs and found a bias towards the cell type where the interacting TEs display enhancer profiles (*Figure 3C*). In contrast, the expression of genes interacting with TE- enhancers was on average similar between the two cell lineages. To test for tissue specificity more generally, we analysed gene expression levels across a wide array of tissues using data from the ENCODE project. Strikingly, genes interacting with TE+ enhancers were, on average, almost exclusively expressed in ESCs or TSCs, whereas those interacting with TE- enhancers displayed high expression across a broad range of tissues (*Figure 3D*). A similar pattern was observed when analysing gene expression at early stages of ESC or TSC differentiation (*Figure 3—figure supplement 2*). These data are in line with the chromatin accessibility data and suggest that TE+ enhancers may be used to support lineage-specific expression of a subset of genes in early development. However, it is also possible for the correlations observed here to emerge in the absence of causal links between TEs and gene expression.

## Genetic editing identifies functional TE-derived enhancers

To directly test for a functional role of TE+ enhancers in gene regulation, we performed CRISPR-mediated genetic excision of selected TEs. Based on the bioinformatic analyses above, we chose a set of 4 TEs in ESCs and 2 TEs in TSCs that display strong evidence of enhancer activity (*Figure 4A, B*; *Figure 4—figure supplement 1*). Most genes targeted by these TE+ enhancers, as predicted from PCHi-C data, had skewed expression towards ESCs or TSCs (*Figure 4D*). We also selected an element interacting with *Smarcad1*, a gene involved in pluripotency maintenance (*Hong et al., 2009*), and one element that was previously shown to regulate *Akap12* in ESCs (*Sundaram et al., 2017*). After genetically excising TE+ enhancers using pairs of sgRNAs (*Figure 4C*) in multiple clones, we measured the effects on the expression of target genes. In TSCs, both TEs tested were found to be key regulators of their predicted target genes, *Map3k8* and *Scarf2*, as homozygous null clones displayed ~4-fold reduction in expression (*Figure 4E,F*). In ESCs, however, only one of the tested TEs had a pronounced effect on the expression of its main target gene (*Figure 4G–J*). Excision of this particular RLTR13D6 element led to a dramatic ~8-fold reduction in *Tdrd12* expression, and also had minor effects on the expression of other nearby genes that interact with it (*Figure 4G*). Although the associated TE lies within the first intron of *Tdrd12*, we found no evidence that it acted as an alternative promoter in ESCs (*Figure 4—figure supplement 2*), supporting its role as an enhancer. Heterozygous deletion of the *Smarcad1*-interacting TE also led to a small but significant decrease in gene expression (*Figure 4H*). It remains unclear whether failure to isolate homozygous clones was due to a loss of ESC self-renewal caused by *Smarcad1* depletion (*Hong et al., 2009*). Excision of the remaining two TEs in ESCs had no effect on the expression of target genes (*Figure 4I,J*), including *Akap12*, suggesting that the previously reported regulatory role of this TE is cell line-dependent (*Sundaram et al., 2017*).

In addition to genetic editing experiments, we also analysed data from an ESC line of a hybrid 129 × Cast background, which displays substantial sequence variation between alleles. Using ATAC-seq data from this line (*Giorgetti et al., 2016*), we first identified 98 TEs with biased chromatin accessibility signal across the two genetic backgrounds, suggestive of allele-specific enhancer activity. This included five elements that were polymorphic between 129 and Cast (according to structural variation data from the Mouse Genomes Project), as well as elements present in both species but bearing sequence variation. We then analysed RNA-seq data from the same cell line (*Gendrel et al., 2014*) to test for effects on allelic gene expression. However, out of 52 genes within 100 kb of a TE-derived allele-specific enhancer, only four displayed a > 1.5 fold difference in expression between alleles (*Figure 4—figure supplement 3*).

Through these genetic approaches we have identified a small set of TEs that play a functional role in the regulation of gene expression in ESCs or TSCs. Yet other TEs play only minor, redundant or no role in gene regulation, despite strong correlative evidence at the level of chromatin composition and conformation.

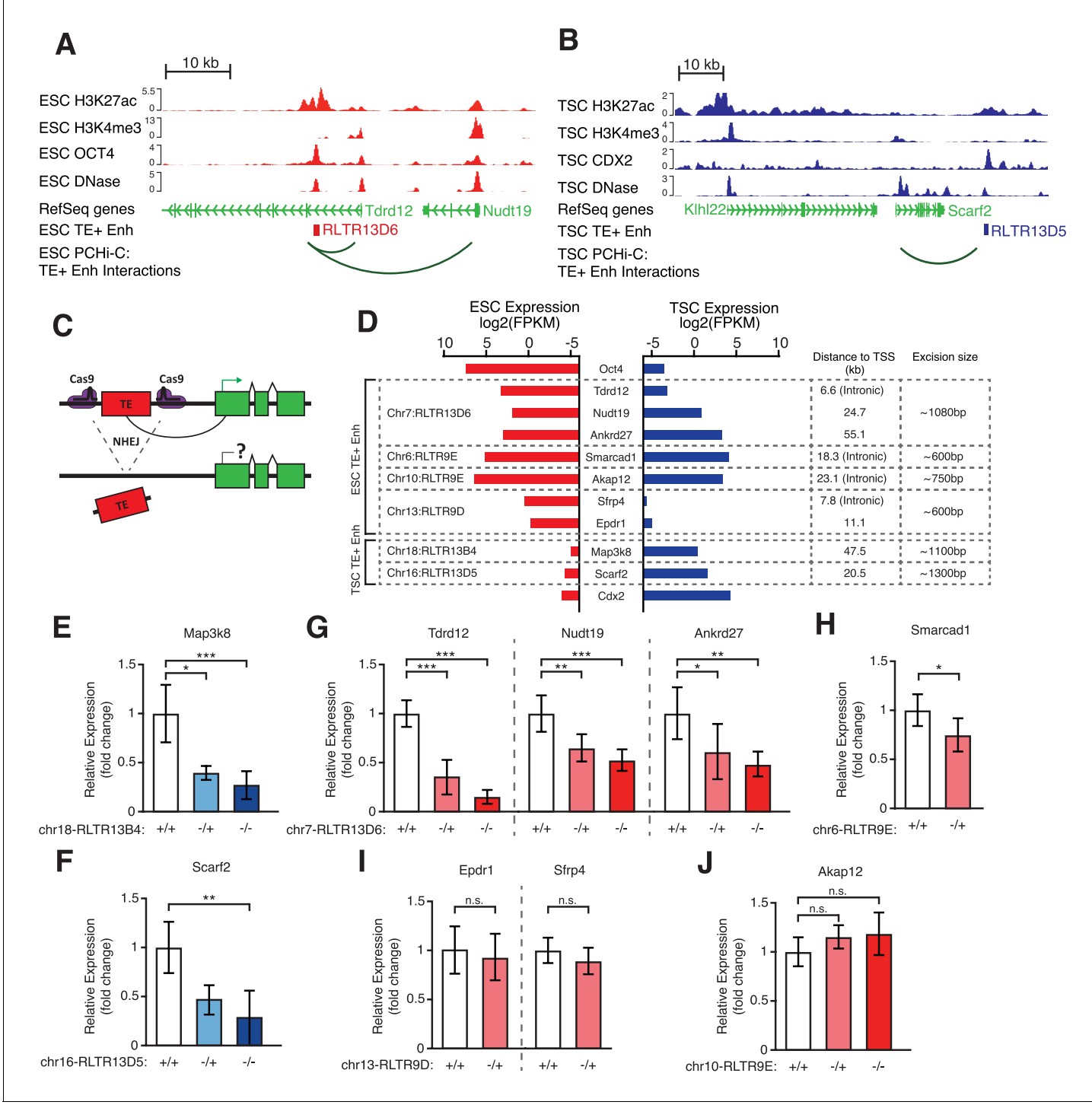

**Figure 4.** Genetic editing identifies regulatory TEs from candidate TE+ enhancers. (**A,B**) Genome browser snapshots showing examples of candidate TE+ enhancers in ESCs (**A**) or TSCs (**B**). ChIP-seq and DNase-seq tracks are displayed, as well as PCHi-C-identified interactions between gene promoters and the respective TE+ enhancer. (**C**) Schematic of the CRISPR strategy for genetically excising TE+ enhancers. (**D**) Expression values from RNA-seq data for the genes interacting with TE+ enhancers that were genetically excised. The expression levels for *Oct4* and *Cdx2* are also displayed as a comparison term. Additional details about the relative location of the TE and the size of the excision are displayed. (**E–J**) RT-qPCR data from TSC (**E–F**) or ESC (**G–J**) clones isolated from CRISPR experiments targeted to the indicated TE+ enhancers. Values are mean ± s.d. from the following number of independent clones: 3 +/+, 2 -/+, 8 -/- (**E**); 3 +/+, 3 -/+, 6 -/- (**F**); 3 +/+, 3 -/+, 4 -/- (**G**); 2 +/+, 2 -/- (**H**); 2 +/+, 2 -/+ (**I**); 2 +/+, 2 -/+, 2 -/- (**J**). For H-J, where clone numbers are low, expression values from three different passage numbers were included for each clone. *p<0.05, **p<0.01, ***p<0.001, ANOVA with Tukey post-hoc test.

*Figure 4 continued on next page*

*Figure 4 continued*

The online version of this article includes the following figure supplement(s) for figure 4:

**Figure supplement 1.** Additional profiles of candidate TE+ enhancers.
**Figure supplement 2.** *Tdrd12* transcriptional profile.
**Figure supplement 3.** Identification of putative TE+ enhancers based on allele-specific data.

## Pan inactivation of RLTR13D6 elements in ESCs reveals minor contribution to gene regulation

To test in a single experiment the regulatory roles of multiple TE+ enhancers in ESCs we performed CRISPR interference (CRISPRi) targeted at RLTR13D6 elements. We designed two sets of 4 sgRNAs each, with distinct strategies in mind (*Figure 5—figure supplement 1A*): a) set 1 maximised the number of RLTR13D6 elements predicted to be targeted by at least one sgRNA (n = 420 elements with no mismatches, out of 805); b) set 2 maximised the number of sgRNAs targeted to each RLTR13D6 element, resulting in a smaller number of copies being targeted (n = 129). The only substantial predicted off-target effects were with TEs of related RLTR13 subfamilies (*Figure 5—figure supplement 1B*), which we took into account for downstream analyses. We then established ESC lines stably expressing a dCas9-KRAB fusion protein, followed by lentiviral transduction of either sgRNA set or an empty vector control. After selection of cells expressing both dCas9-KRAB and sgRNAs, we performed H3K27ac ChIP-seq and RNA-seq analyses 5–8 days post-infection. Although we failed to get adequate signal from a Cas9 ChIP-seq, analysis of published data from a CRISPRi experiment on human LTR5Hs elements (*Fuentes et al., 2018*) showed that the number of sgRNAs predicted to bind each element strongly correlates with Cas9 ChIP-seq signal (*Figure 5—figure supplement 1C*). As 97% of Cas9 binding events at LTR5Hs elements could be predicted in silico, we used sgRNA binding predictions as a surrogate measure for Cas9 binding.

Analysis of the H3K27ac ChIP-seq data revealed a reduction in H3K27ac signal at RLTR13D6 elements that were predicted to be targeted by each of the sgRNA sets, whereas H3K27ac levels at RLTR9 elements were unaffected (*Figure 5A,B*). Notably, elements targeted by multiple sgRNAs displayed a larger reduction in H3K27ac than those targeted by a single sgRNA (*Figure 5A,B*), in line with recent findings (*Fuentes et al., 2018*). For sgRNA set 1, CRISPRi resulted in a > 2 fold loss of H3K27ac signal at 30 (56%) RLTR13D6-targeted elements that overlapped a H3K27ac peak. A similar number was obtained for sgRNA set 2 (n = 25, 46%), with 34 elements being affected in total across the two sets (out of 76 H3K27ac-marked elements). We then asked how these changes affected gene expression. Strikingly, only three genes were significantly differentially expressed upon CRISPRi across both sets of sgRNAs: *Tdrd12*, *Spp1* and *Hook3* (*Figure 5C,D*). All three cases were associated with a RLTR13D6 element targeted by one or both of the sgRNA sets (*Figure 5E–G*). Most likely these elements act as distal enhancers given that they lay >6 kb away from the respective transcriptional start sites, whereas heterochromatin spreading as a result of CRISPRi tends to be limited to the open chromatin regions targeted by the sgRNAs (*Thakore et al., 2015*; *Fuentes et al., 2018*). *Tdrd12* stood out as the most downregulated gene after CRISPRi with either sgRNA set, which together with results from genetic editing experiments above (*Figure 4G*) provide ample evidence that *Tdrd12* expression in ESCs critically depends on an intronic RLTR13D6 element. Silencing of this element did not cause any structural alterations of the *Tdrd12* transcript, supporting its role as an enhancer (*Figure 4—figure supplement 2*). The effect of CRISPRi on *Hook3* expression was far more subtle, despite a pronounced loss of H3K27ac at the associated RLTR13D6 element (*Figure 5G*). Notably, *Spp1* downregulation occurred in a manner that seemed largely independent of changes in H3K27ac levels (*Figure 5F*), suggesting that enhancer inactivation occurred through deposition of repressive marks and/or via impairment of TF binding by dCas9-KRAB. For the remainder of the targeted TEs, although as a group the associated genes displayed a significant downregulation upon CRISPRi, these changes were limited to at most 1.4-fold (*Figure 5—figure supplement 2*). This included genes interacting with TE+ enhancers displaying a similar or greater H3K27ac loss to that observed at *Tdrd12*- or *Spp1*-associated RLTR13D6 elements. To validate the relatively small effect of some TE+ enhancers on gene expression, we performed additional genetic editing experiments. We chose two RLTR13D6 elements that displayed >2-fold change in H3K27ac levels upon CRISPRi (*Figure 5—figure supplement 3A,D*), but with little or no effect on neighbouring gene

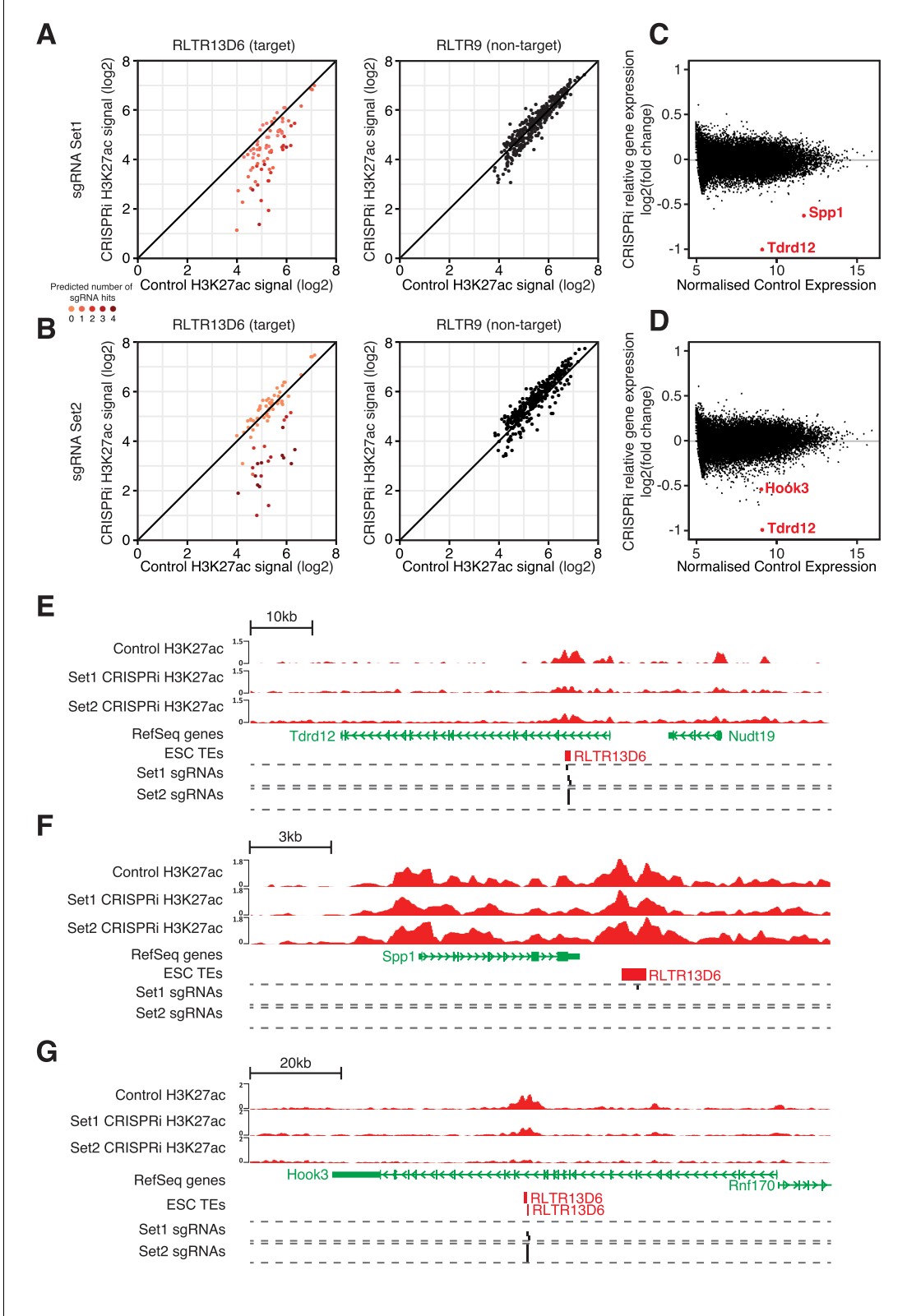

**Figure 5.** CRISPR interference reveals minor contribution of RLTR13D6 elements to ESC gene regulation. (A,B) Normalised H3K27ac signal in dCas9-KRAB-expressing ESCs transduced with sgRNA set 1 (A) or set 2 (B) targeted to RLTR13D6 elements, or with empty vector. Data shown are for elements that overlap H3K27ac peaks in control cells. Colour code indicates the number of predicted sgRNA hits for each element. Data for RLTR9 are also shown as a control, non-targeted group. (C,D) MA plot for the RNA-seq data from triplicate CRISPRi experiments using sgRNA set 1 (C) or set 2 (D).

*Figure 5 continued on next page*

*Figure 5 continued*

Genes with statistically significant differential expression are highlighted in red. (**E–F**) Genome browser snapshots for the genes highlighted in C,D. H3K27ac ChIP-seq tracks for control or CRISPRi ESCs are shown, as well as predicted sgRNA hits from each set.

The online version of this article includes the following figure supplement(s) for figure 5:

**Figure supplement 1.** Target predictions for sgRNAs used for CRISPRi.
**Figure supplement 2.** Effects of CRISPRi on H3K27ac levels and gene expression.
**Figure supplement 3.** Genetic editing of TEs with little or no effect on gene expression upon CRISPRi.

expression (*Figure 5—figure supplement 3B,E*). Upon CRISPR-mediated deletion of the respective TEs, gene expression levels largely agreed with CRISPRi results, with only the lowly expressed *Tsacc* gene being downregulated, although the effect was more pronounced in the genetic excision experiment (*Figure 5—figure supplement 3C,F*). Therefore, whilst CRISPR of individual TEs constitutes a more sensitive approach, our CRISPRi data correlates with genetic excision data and is robust with respect to large effects on gene expression.

These results suggest that only a small minority of RLTR13D6 elements play a major role in the regulation of gene expression in ESCs. This is in contrast with the broader correlations found by analysis of epigenomic and transcriptomic data in wildtype cells, highlighting the need to establish causal roles via direct molecular manipulation of TEs.

## Discussion

TEs are increasingly being presented as major contributors to gene regulatory networks in a variety of contexts (*Chuong et al., 2013*; *Chuong et al., 2016*; *Kunarso et al., 2010*; *Jacques et al., 2013*; *Sundaram et al., 2014*; *Lynch et al., 2011*; *Fuentes et al., 2018*; *Imbeault et al., 2017*). Yet most of these studies have relied largely on the idea that biochemical activity at the chromatin level is indicative of function, a concept famously associated with the findings of the ENCODE project (*ENCODE Project Consortium, 2012*) that triggered a still ongoing debate (*Graur et al., 2013*; *Ponting, 2017*; *Doolittle and Brunet, 2017*). The use of genetic and epigenetic editing tools as presented here, and also used by other labs (*Chuong et al., 2016*; *Jachowicz et al., 2017*; *Fuentes et al., 2018*), initiate a much needed move to evaluating causal roles for TEs in gene regulation.

Our work has revealed that a set of TEs with regulatory potential in ESCs act mostly neutrally with respect to their effects on gene expression, which contrasts with earlier suggestions from analyses of chromatin profiling experiments (*Kunarso et al., 2010*). In the absence of a substantial contribution of ESC TE+ enhancers to gene regulation, the striking correlations emerging from profiling efforts most likely reflect the fact that TE insertions are best tolerated in regions where their tissue-specific enhancer action matches the expression profiles of nearby genes. Whilst the enhancer activity of these TEs could be inconsequential for gene expression, this genomic 'safe niche' would be permissive for fixation by genetic drift. Additionally, we cannot exclude the possibility that some TE + enhancers act redundantly with TE- enhancers, despite our attempt to isolate the effects of each of these enhancer groups. Such TEs could still be important to ensure regulatory robustness and, indeed, enhancer redundancy is a seemingly common feature of enhancer-gene networks (*Osterwalder et al., 2018*). Extensive genetic work will be necessary to evaluate what proportion of TE+ enhancers act in a redundant fashion with other regulatory elements.

In a contrasting example to our findings, epigenetic editing work by the Wysocka lab has revealed that a large proportion of LTR5Hs elements play significant roles in the regulation of nearly 300 genes in a human embryonal carcinoma cell line (*Fuentes et al., 2018*). Notably, in the latter study virtually all H3K27ac-marked LTR5Hs elements were inactivated by CRISPRi (257 copies, according to our own analysis), probably due to the use of twelve sgRNAs simultaneously (*Fuentes et al., 2018*). It is therefore possible that more efficient targeting of all H3K27ac-marked RLTR13D6 elements (76 copies) in mouse ESCs would uncover additional regulatory elements. However, the effects seen here from silencing 34 (44%) of these copies suggest that regulatory RLTR13D6 elements would likely remain a minority. Another explanation to the differences between LTR5Hs and RLTR13D6 action is that the regulatory effects of TEs are variable between families and cellular contexts. Indeed, other TEs considered here and whose effects we did not test by CRISPRi

(RLTR9, RLTR13D5, RLTR13B), may play important roles in ESC and TSC gene regulation. These considerations further emphasise the need to perform functional experiments on a case-by-case basis.

Despite the neutral action of most TEs analysed here, we have uncovered a number of elements that act as key enhancers of gene expression in ESCs (*Tdrd12*, *Smarcad1*, *Spp1* and *Hook3*) and TSCs (*Map3k8* and *Scarf2*). But do these TE insertions impact on cellular and organismal phenotypes, ultimately affecting host fitness? TDRD12 is a protein essential for secondary piRNA production in mice and is essential for male fertility (*Pandey et al., 2013*; *Pandey et al., 2018*). It is therefore possible that the RLTR13D6 element we identified is also active during male germ line development, which would ironically implicate it in genome defence against the mobility of younger, piRNA-targeted TEs. *Smarcad1* knockout mice are subviable, displaying growth defects and low fertility (*Schoor et al., 1999*). It remains to be seen whether the activity of the RLTR9E element we identified plays any role in these phenotypes, possibly by affecting early embryonic differentiation (*Hong et al., 2009*). Other genes that we found to be regulated by TEs are not essential for development: both *Map3k8* (*Dumitru et al., 2000*) or *Spp1* (*Rittling et al., 1998*) knockout mice are viable and appear to develop normally. In these cases it is likely that the action of the associated TEs is inconsequential to the organism, although there could be more subtle embryonic phenotypes or these TEs could play additional roles outside of the early developmental context. Notably, most exemplars of phenotypically relevant TE-derived regulatory elements have been uncovered by analysing naturally occurring phenotypes (*Lisch, 2013*; *Chuong et al., 2017*). The reverse approach of searching for phenotypes linked with putative *cis*-acting TEs has the potential to reveal a wide array of adaptive TE insertions, although this is nonetheless challenging and examples to date are limited (*Chuong et al., 2017*).

At a time when epigenomic data are providing abundant indications that TEs play functional regulatory roles, our findings place these observations into perspective and provide a reminder that a large proportion of mammalian genomic sequences are neutrally evolving. Yet evolutionary tinkering (*Jacob, 1977*) may still benefit from a large amount of 'junk DNA' that occasionally can be put to good use. As was fittingly put by Goodier and Kazazian, 'evolution has been adept of turning some 'junk' into treasure' (*Goodier and Kazazian, 2008*).

# Materials and methods

## Cell culture

E14 ESCs (ATCC CRL-1821) were grown in feeder-free conditions in DMEM GlutaMAX medium (Thermo Fisher) supplemented with 15% FBS, non-essential amino acids, 50 µM 2-mercaproethanol and 1,000 U/ml ESGRO LIF (Millipore). TS-Rs26 cells (a kind gift from Dr. Myriam Hemberger) were cultured under routine conditions (*Tanaka et al., 1998*): 20% fetal bovine serum, 1 mM Na-pyruvate, Pen/Strep, 50 µM 2-mercaproethanol, 25 ng/ml bFGF and 1 µg/ml heparin in RPMI1640, with 70% of the medium pre-conditioned on embryonic feeder cells. The identity of the cells was confirmed through RT-qPCR analyses of key ESC and TSC expression markers. Cells were negative for mycoplasma, as tested using the Sigma LookOut PCR kit.

## TE excisions by CRISPR-Cas9

Two sgRNA sequences flanking each TE of interest were designed with the use of the Zhang lab online tool (http://cripsr.mit.edu/) and cloned into pSpCas9(BB)−2A-GFP (Addgene #48138) (*Ran et al., 2013*) or a modified version of the eSpCas9(1.1) plasmid system (Addgene #71814) (*Slaymaker et al., 2016*). The pSpCas9/eSpCas9 plasmid constructs were transfected into ESCs and TSCs using FuGENE 6 (Promega) with 4 µg equimolar mix of 5' sgRNA and 3' sgRNA in 6-well plates. Single GFP-positive cells were sorted into 96-well plates 48 hr post-transfection. After 7–10 days, growing colonies were genotyped using DNA isolated using QuickExtract (Lucigen) and the primers listed on *Supplementary file 2*. Selected clones were grown further into 6-well plates before collecting RNA using QIAGEN's DNA/RNA mini kit or QIAzol reagent. For some clones, multiple RNA collections were performed at different passages (see legend to *Figure 4*). RNA was DNAse treated with the TURBO DNA-free Kit (Ambion) and reverse transcribed with the RevertAid First Strand cDNA Synthesis kit (Thermo Fisher) using supplier protocol with 100 ng - 2 µg input RNA.

The MESA BLUE SYBR Green mastermix (Eurogentec) was used for qPCR analysis (primers listed on *Supplementary file 2*).

## CRISPRi

To generate stable ESC line expressing dCas9-KRAB, cells were infected with a lentiviral vector (a kind gift from Mark Dawson, Peter MacCallum Cancer Centre, Melbourne, Australia) and mCherry-positive cells were sorted 48–96 hr post-infection. Guide RNA sequences targeting retroelement classes were designed with an R script which downloads all relevant retroelement sequences, identifies potential guide sequences within each sequence and returns guide candidates which match the highest number of targets. Potential off-target effects were verified using the Cas-OFFinder tool (*Bae et al., 2014*) and sgRNAs with minimal off-targets were selected. Designed sgRNA sequences were cloned into a lentiviral sgRNA vector (a kind gift from Mark Dawson) and packaged into viral particles. The dCas9-KRAB-expressing ESCs were infected and cells sorted at 48 hr to collect dual positive mCherry/BFP positive cells. Cells infected with the empty sgRNA vector were used as a control. RNA was collected by QIAzol extraction at 5 days or 8 days post-infection of the sgRNA lentivirus. Chromatin was collected at 8 days post-infection.

## RNA-seq

Ribosomal RNA-depleted RNA-seq libraries were prepared with the NEBNext rRNA Depletion Kit (New England BioLabs) from 400 to 600 ng of QIAzol extracted total RNA. Libraries were sequenced on an Illumina NextSeq 500 with single-ended 75 bp reads at the Barts London Genome Centre.

## ChIP-seq

Cells were fixed with 1% formaldehyde in PBS for 12 min, which was then quenched with glycine (final concentration 0.125 M). Fixed cells were washed and lysed as previously described (*Latos et al., 2015*). Chromatin was sonicated to an average size of 200–700 bp using a Bioruptor Pico (Diagenode). Immunoprecipitation was performed using 15 µg of chromatin and 2.5 µg of anti-H3K27ac antibody (Active Motif #39133). DNA purification was performed using the GeneJET PCR Purification Kit (Thermo Fisher) with DNA eluted in 80 µL of elution buffer. ChIP-seq libraries were prepared from 1 to 5 ng eluted DNA using NEBNext Ultra II DNA library Prep Kit (New England BioLabs). Libraries were sequenced on an Illumina NextSeq 500 with single-ended 75 bp reads at the Barts London Genome Centre.

## Primary data processing

ChIP-seq and ATAC-seq data generated here or from external datasets (*Supplementary file 1*) were mapped by trimming reads using Trim_galore! and aligning to the mm10 genome assembly using Bowtie2 v2.1.0 (*Langmead and Salzberg, 2012*), followed by filtering of uniquely mapped reads. Data were normalised to total read count. ChIP-seq peak detection was performed using MACS2 v2.1.1 (*Zhang et al., 2008*) with -q 0.05; for histone marks the option `–broad` was used. ATAC-seq peak detection was performed using F-seq v1.84 (*Boyle et al., 2008*) with options `–f 0 t 6`. For multi-tissue DNAse-seq data, peak annotation files generated by ENCODE were used.

RNA-seq data generated here or from external datasets (*Supplementary file 1*) were mapped by trimming reads using Trim_galore! and aligning to the mm10 genome assembly with Tophat v2.0.9 (*Trapnell et al., 2009*) using a transcriptome index from Illumina's iGenomes. For ENCODE multi-tissue RNA-seq data, FPKM expression values were downloaded directly from ENCODE and the data were normalised by histogram matching.

Processed CpG calls from publicly available BS-seq and TAB-seq data were downloaded from the respective GEO submissions (*Supplementary file 1*).

## TE and enhancer annotations

To identify TE+ enhancers, coordinates for RLTR9, RLTR13D6, RLTR13D5 and RLTR13B elements were taken from the mm10 RepeatMasker annotation and filtered to remove elements either intersecting H3K4me3 ChIP-seq peaks or lying within 500 bp of a TSS from the NCBI RefSeq annotation. Enhancer-like elements were then selected if they intersected with all three of the following:

H3K27ac ChIP-seq peaks, ATAC-seq peaks and binding sites for any of three key TFs (OCT4, NANOG or SOX2 for ESC TE+ enhancers; CDX2, ELF5 or EOMES for TSC TE+ enhancers).

To define TE- enhancers, ATAC-seq peaks that did not overlap any TEs annotated by Repeat-Masker were used as a basis for potential enhancer regions. These regions were then filtered in the same manner as described for TE+ enhancers.

Non-enhancer TEs were defined as mappable elements displaying low ATAC-seq signal. Mappability scores were obtained by mapping in silico generated reads and measuring the proportion of the element's length covered by uniquely mapped reads. Elements with a score higher than 0.5 were kept and from those with the lowest ATAC-seq signal selected as non-enhancer TEs. The number of elements selected for each class equated to twice the number of TE+ enhancers identified for the same class.

### ChIP-seq simulations

To estimate how many ChIP-seq peaks could be missed due to low read mappability to TEs we used ChIPulate (*Datta et al., 2019*) to simulate ChIP-seq data. We generated reads for simulated peaks centred on each TE of interest, as well as control reads spanning 10 kb around each peak, with the following parameters: `–read-length 40 –fragment-length 150 –fragment-jitter 40`. These reads were mapped back to the mouse genome followed by peak detection using MACS2 as described above. As a control, peak detection was also performed on the simulated (pre-mapping) data, which successfully detected all TE copies.

### Linking TEs and enhancers to gene promoters

Processed promoter-genome spatial interactions were downloaded from the respective ArrayExpress submission (*Supplementary file 1*). Each element of interest (TE+ enhancers, TE- enhancers, non-enhancer TEs) was intersected with the list of non-promoter restriction fragments in the PCHi-C data and coupled to the gene promoter(s) it interacted with. Expression values from RNA-seq data were assigned to each element based on these relationships. To distinguish the putative effects of TE+ and TE- enhancers, only genes interacting exclusively with one type of enhancer were considered.

### Motif analysis

Motif analysis of TE copies was performed using the FIMO tool of the MEME SUITE v5.0.1 (*Bailey et al., 2015*) using the HOCOMOCO v11 TF motif database. Data from the FIREWACh enhancer reporter assay were obtained from the respective publication (*Murtha et al., 2014*). Briefly, FIREWACh uses a library of restriction-digested DNA fragments from accessible regions of the genome, which are cloned into a lentiviral plasmid containing a GFP reporter. The lentiviral library was transduced into ESCs, followed by sorting of GFP+ cells and sequencing of the cloned fragments. For our analysis, we used the coordinates of both the identified regulatory elements and the input library fragments.

### Hybrid 129 × Cast ESC data

Processed allele-specific data from ATAC-seq experiments on 129 × Cast hybrid ESCs were downloaded from the respective GEO submission (*Supplementary file 1*). Peaks containing at least five reads in one of the alleles and an allelic ratio (129 reads/total reads) larger than 0.8 or lower than 0.2 were selected as allele-specific regulatory elements. These were intersected with TE annotations to identify putative allele-specific TE-derived regulatory elements. RNA-seq data from the same cell line (*Supplementary file 1*) were used to extract expression values for genes within 100 kb of allele-specific TE regulatory elements.

## Acknowledgements

We thank Myriam Hemberger and Vicente Perez-Garcia for help with setting up CRISPR, Mark Dawson for CRISPRi plasmids, Neza Vadnjal and Maki Iwai for help with plasmid cloning, Vasavi Sundaram for comments on the manuscript, and Wolf Reik for kindly hosting experiments in his lab at the revision stage.

## Additional information

### Funding

| Funder | Grant reference number | Author |
|---|---|---|
| Wellcome | Sir Henry Dale Fellowship 101225/Z/13/Z | Miguel R Branco |
| The Medical College of Saint Bartholomew's Hospital Trust | Donald Hunter Studentship | Christopher D Todd |
| Biotechnology and Biological Sciences Research Council | BB/R505997/1 | Darren Taylor |

The funders had no role in study design, data collection and interpretation, or the decision to submit the work for publication.

### Author contributions

Christopher D Todd, Formal analysis, Investigation, Visualization, Methodology, Writing—review and editing, Designed experiments, Performed cell culture, CRISPR, CRISPRi, RNA-seq and bioinformatic analyses, Wrote the manuscript; Özgen Deniz, Investigation, Methodology, Writing—review and editing, Performed ChIP-seq, Provided feedback on the manuscript; Darren Taylor, Investigation, Assisted with CRISPR experiments; Miguel R Branco, Conceptualization, Formal analysis, Supervision, Funding acquisition, Methodology, Writing—original draft, Project administration, Designed the study and experiments, Performed bioinformatic analyses, Wrote the manuscript

### Author ORCIDs

Christopher D Todd (ID) https://orcid.org/0000-0003-2663-6173
Özgen Deniz (ID) https://orcid.org/0000-0001-7268-1923
Miguel R Branco (ID) https://orcid.org/0000-0001-9447-1548

### Decision letter and Author response

Decision letter https://doi.org/10.7554/eLife.44344.sa1
Author response https://doi.org/10.7554/eLife.44344.sa2

## Additional files

### Supplementary files

• Supplementary file 1. List of external datasets used.

• Supplementary file 2. List of primers and sgRNAs used.

• Transparent reporting form

### Data availability

ChIP-seq and RNA-seq data generated in this study are available from the NCBI Gene Expression Omnibus repository under the accession number GSE122856. Sources of external datasets used are detailed in Supplementary file 1. Scripts used for data analysis are available from Github (https://github.com/Christopher-Todd/Todd-eLife-2019; copy archived at https://github.com/elifesciences-publications/Todd-eLife-2019).

The following dataset was generated:

| Author(s) | Year | Dataset title | Dataset URL | Database and Identifier |
|---|---|---|---|---|
| Todd CD, Deniz O, Branco MR | 2018 | Functional evaluation of transposable elements as transcriptional enhancers in mouse embryonic and trophoblast stem cells | https://www.ncbi.nlm.nih.gov/geo/query/acc.cgi?acc=GSE122856 | NCBI Gene Expression Omnibus, GSE122856 |

The following previously published datasets were used:

| Author(s) | Year | Dataset title | Dataset URL | Database and Identifier |
|---|---|---|---|---|
| Mouse ENCODE Consortium | 2013 | A comparative encyclopedia of DNA elements in the mouse genome | https://www.ncbi.nlm.nih.gov/geo/query/acc.cgi?acc=GSE49847 | NCBI Gene Expression Omnibus, GSE49847 |
| Newman JJ, Bilodeau S, Mullen AC, Reddy J, Whyte W, Orlando D, Abraham BJ, Hnisz D, Young RA | 2013 | Master Transcription Factors and Mediator Establish Super-Enhancers at Key Cell Identity Genes | https://www.ncbi.nlm.nih.gov/geo/query/acc.cgi?acc=GSE44288 | NCBI Gene Expression Omnibus, GSE44288 |
| iefke R, Karwacki-Neisius V, Shi Y | 2016 | EPOP interacts with Elongin BC and USP7 to modulate the chromatin landscape | https://www.ncbi.nlm.nih.gov/geo/query/acc.cgi?acc=GSE90045 | NCBI Gene Expression Omnibus, GSE90045 |
| Shen Y, Yue F, Ren B | 2012 | A draft map of cis-regulatory sequences in the mouse genome | https://www.ncbi.nlm.nih.gov/geo/query/acc.cgi?acc=GSE29184 | NCBI Gene Expression Omnibus, GSE29184 |
| Kagey MH, Bilodeau S, Newman JJ, Orlando DA, Young RA | 2010 | Control of Embryonic Stem Cell State by Mediator and Cohesin | https://www.ncbi.nlm.nih.gov/geo/query/acc.cgi?acc=GSE22557 | NCBI Gene Expression Omnibus, GSE22557 |
| Chuong EB, Baker JC | 2012 | Rodent trophoblast epigenome | https://www.ncbi.nlm.nih.gov/geo/query/acc.cgi?acc=GSE42207 | NCBI Gene Expression Omnibus, GSE42207 |
| Domcke S, Bardet AF, Schübeler D | 2015 | Competition between DNA methylation and transcription factors determines binding of NRF1 | https://www.ncbi.nlm.nih.gov/geo/query/acc.cgi?acc=GSE67867 | NCBI Gene Expression Omnibus, GSE67867 |
| Wu J, Huang B, Chen H, Xie W | 2016 | The landscape of accessible chromatin in mammalian pre-implantation embryos | https://www.ncbi.nlm.nih.gov/geo/query/acc.cgi?acc=GSE66390 | NCBI Gene Expression Omnibus, GSE66390 |
| Nelson A, Mould A, Bikoff E, Robertson E | 2017 | ATAC-seq analysis of the chromatin landscape during in vitro differentiation of murine trophoblast stem cells | https://www.ncbi.nlm.nih.gov/geo/query/acc.cgi?acc=GSE94694 | NCBI Gene Expression Omnibus, GSE94694 |
| Smith ZD, Shi J, Dongahey J, Cacciarelli D, Michor F, Meissner A | 2017 | Epigenetic restriction of embryonic and extraembryonic lineages mirrors the somatic transition to cancer | https://www.ncbi.nlm.nih.gov/geo/query/acc.cgi?acc=GSE84236 | NCBI Gene Expression Omnibus, GSE84236 |
| Carter AC, Xu J, Chang HY | 2016 | Allele-specific ATAC-seq | https://www.ncbi.nlm.nih.gov/geo/query/acc.cgi?acc=GSE71156 | NCBI Gene Expression Omnibus, GSE71156 |
| Calabrese JM | 2012 | Site-specific silencing of regulatory elements as a mechanism of X-inactivation | https://www.ncbi.nlm.nih.gov/geo/query/acc.cgi?acc=GSE39406 | NCBI Gene Expression Omnibus, GSE39406 |
| Kim H | 2018 | Prerequisite Barcoding of Cell-Type-Restricted Enhancers by ESC Transcription Factors in ESCs Licenses Their Robust Developmental Activation | https://www.ncbi.nlm.nih.gov/geo/query/acc.cgi?acc=GSE81681 | NCBI Gene Expression Omnibus, GSE81681 |
| Hon G, Ren B | 2014 | 5mC Oxidation by Tet2 Modulates Enhancer Activity and Timing of Transcriptome Reprogramming during Differentiation | https://www.ncbi.nlm.nih.gov/geo/query/acc.cgi?acc=GSE48519 | NCBI Gene Expression Omnibus, GSE48519 |
| Cambuli F, Murray A, Dean W, Dudzinska D, Krueger F, Andrews S, Senner CE, Cook SJ, Hemberger M | 2014 | Epigenetic memory of the first cell fate decision prevents complete ES cell reprogramming into trophoblast | https://www.ncbi.nlm.nih.gov/geo/query/acc.cgi?acc=GSE62150 | NCBI Gene Expression Omnibus, GSE62150 |
| Latos P, Sienerth A, Murray A, Senner C, Muto M, Ikawa M, Oxley D, Burge | 2015 | Transcriptional regulation of trophoblast cell fate | https://www.ebi.ac.uk/ena/data/view/PRJNA298763 | EBI European Nucleotide Archive, PRJNA298763 |

| | | | | |
|---|---|---|---|---|
| S, Cox B, Hemberger M | | | | |
| Gendrel A, Attia M, Chen C, Diabangouaya P, Servant N, Barillot E, Heard E | 2014 | The developmental dynamics and disease potential of random monoallelic gene expression | https://www.ncbi.nlm.nih.gov/geo/query/acc.cgi?acc=GSE54016 | NCBI Gene Expression Omnibus, GSE54016 |
| Mifsud B, Branco M | 2018 | Promoter Capture Hi-C of mouse ESC and TSC | https://www.ebi.ac.uk/arrayexpress/experiments/E-MTAB-6585/ | EBI ArrayExpress, E-MTAB-6585 |
| Bonev B, Mendelson Cohen N, Szabo Q, Fritsch L, Papadopoulos G, Lubling Y, Xu X | 2017 | Multi-scale 3D genome rewiring during mouse neural development | https://www.ncbi.nlm.nih.gov/geo/query/acc.cgi?acc=GSE96107 | NCBI Gene Expression Omnibus, GSE96107 |

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
