## [Decision Letter]

Thank you for submitting your article "Functional evaluation of transposable elements as enhancers in mouse embryonic and trophoblast stem cells" for consideration by *eLife*. Your article has been reviewed by three peer reviewers, including Deborah Bourc'his as the Reviewing Editor and Reviewer #1, and the evaluation has been overseen by Detlef Weigel as the Senior Editor. The following individuals involved in review of your submission have also agreed to reveal their identity: Alvaro Rada-Iglesias (Reviewer #2); Dixie Mager (Reviewer #3).

The reviewers have discussed the reviews with one another and the Reviewing Editor has drafted this decision to help you prepare a revised submission.

Summary:

Transposable elements (TEs) are more and more postulated to act as regulatory sequences that could drive the expression of gene networks during development, acting as alternative promoters or enhancers. Genome-wide mapping studies have indeed documented that TEs have often been co-opted as binding sites for transcription factors and can be enriched for chromatin features of enhancers -such as H3K27ac and H3K4me1- in tissue- or stage-specific contexts. However, these proposed regulatory functions are largely based on correlative observations. Using CRISPRi and multiple sgRNA (CARGO) delivery, a recent publication was the first to provide potential functional significance, demonstrating that LTR5HS elements could act as enhancers of a few hundred genes in the human genome. Regarding the variety of transposable elements in terms of evolutionary history, sequences, numbers and multiplication success, more studies are clearly needed to analyze the potential of retrotransposons as developmental regulators of gene expression.

Here the authors focused on ERVK elements that were previously shown to display enhancer features in mouse ESCs (RLTR9 and RLTR13D6) and TSCs (RLTR13B and RLTR13D5). Using available genome-wide datasets (TF occupancy, histone marks, promoter capture), they first defined a set of 684 ESC-specific and 358 TSC-specific enhancer-like LTR elements, along with the gene promoters they may interact with. Using both genetic deletion of a few elements and genome-wide CRISPRi approaches, they observed very limited influence of these LTR elements on enhancing gene expression, and concluded that insertion sites of these elements were probably selected for their expression patterns similar to -and therefore not detrimental to- nearby genes. The conclusions have clear interests, for both the transposon and the enhancer fields, the study is overall well conducted and the manuscript well written. However, according to the three reviewers, it is necessary to better define why this category of LTR elements does not have widespread enhancer properties, despite showing enhancer features: technical reasons (CRISPRi efficiency)? Biological reasons (enhancer redundancy)? Or indeed, lack of functionality?

Essential revisions:

1) It would be helpful to provide a more information about the LTR families chosen for analysis. Why were they chosen? What proviral families are they related to (if any), what are their approximate copy numbers, etc. Admittedly, nothing is known for most such LTR families but anything the authors can provide would make these entities seem less obscure. On the same matter, it is unclear from the beginning whether the authors focused on solo LTRs or full-length elements or both. Please specify. Notably, are there differences between TE+ enhancers and TE non-enhancers regarding element size? Or between TE+ enhancers that are neutral upon CRISPRi and the few that are impactful?

2) When using targeted genetic deletions, the authors found that a fair fraction of LTR elements had effects on gene expression. Thus, the dominant lack of effects when using CRISPRi might be due to partially abrogated enhancer function. Accordingly, the level of H3K27ac downregulation upon CRISPRi appears weak in some instances. Is a 2-fold reduction in H3K27ac enough to lose enhancer properties? The authors should select a couple of transposable elements that according to the CRISPRi had no effects on gene expression and test whether this is also true upon genetic deletion. When those elements that were already studied by both genetic deletion and CRISPRi, the effects on gene expression should be systematically compared between the two approaches.

3) The lack of effects on gene expression is not an exclusive property of transposable elements but of enhancers in general when analyzed individually (this is something that the authors should comment on). As the authors state in their Discussion, one possible explanation is the prevalent enhancer redundancy that characterizes the regulation of many genes. The authors should look for any distinguishing genomic features surrounding the few LTRs that indeed did act as enhancers (ATAC-seq, p300, and promoter HiC data (and others). One might predict that LTR copies with significant impact on gene expression might be found in locations with no other host genome enhancers in the vicinity to serve the same function. In other words, perhaps TEs that have truly become important for gene expression have done so once the natural enhancer(s) for the gene have degraded.

The authors should then functionally test whether this could explain the lack of relevance at least for one locus. Briefly, they could select one gene whose expression is predicted to be controlled by at least two enhancers with one of them being an LTR element here studied. Then the authors could delete each of those enhancers either individually or together and evaluate gene expression changes.

---

## [Author Response]

Essential revisions:1) It would be helpful to provide a more information about the LTR families chosen for analysis. Why were they chosen? What proviral families are they related to (if any), what are their approximate copy numbers, etc. Admittedly, nothing is known for most such LTR families but anything the authors can provide would make these entities seem less obscure. On the same matter, it is unclear from the beginning whether the authors focused on solo LTRs or full-length elements or both. Please specify. Notably, are there differences between TE+ enhancers and TE non-enhancers regarding element size? Or between TE+ enhancers that are neutral upon CRISPRi and the few that are impactful?

We now provide additional information and context about the TE families that were selected for this study, including a new figure (Figure 1—figure supplement 1).

These families were chosen because, based on previously published data, they were the most highly enriched for binding of key transcription factors in each of the cell types. We have clarified this at the start of the Results section: “…we focused on a set of TE families that were previously shown to be highly enriched for binding of key TFs in ESCs (RLTR9, RLTR13D6) (Kunarso et al., 2010) or TSCs (RLTR13B, RLTR13D5) (Chuong et al., 2013)”. As the reviewers correctly point out, not much is known about these LTR families. Within the Repbase database, only two families (RLTR9C and RLTR9E) are directly coupled to internal portions (MMERVK9C-int and MMERVK9E-int). Additional insights on proviral families would require extensive genomic and phylogenetic characterization, which falls beyond the scope of this study.

Using Repeatmasker annotations, we now provide copy numbers for each subfamily considered in this study, as well as an indication of their age based on abundance in different Muridae species (Figure 1—figure supplement 1A, B). We have also analysed the genomic arrangement of LTR pairs to look for suggestive evidence of proviral insertions. Our analysis shows that the vast majority of elements are solo LTRs, whilst only a minority harbours full-length LTRs in the same orientation within 10kb of each other (Figure 1—figure supplement 1C). Both forms (solo LTRs and proviral) were considered in our study. This discussion was added to the text (subsection “TE-derived enhancers in ESCs and TSCs are highly tissue-specific”, first paragraph).

Regarding element length, we have added the following: “TE+ enhancers are on average longer than non-enhancer TEs, as expected if deletions remove key TF binding sites (Figure 2A, B; Figure 2—figure supplement 1B). Nevertheless, there are many non-enhancer TEs that are full-length LTRs, prompting the question of what are the sequence determinants of enhancer activity”. It is important to note that for motif analyses we only considered long LTRs (>60% of maximum length for each family), to try and pinpoint base-level differences that drive enhancer activity. Unfortunately, the comparison between neutral and CRISPRi-modulated TEs cannot be performed due to lack of statistical power.

2) When using targeted genetic deletions, the authors found that a fair fraction of LTR elements had effects on gene expression. Thus, the dominant lack of effects when using CRISPRi might be due to partially abrogated enhancer function. Accordingly, the level of H3K27ac downregulation upon CRISPRi appears weak in some instances. Is a 2-fold reduction in H3K27ac enough to lose enhancer properties? The authors should select a couple of transposable elements that according to the CRISPRi had no effects on gene expression and test whether this is also true upon genetic deletion. When those elements that were already studied by both genetic deletion and CRISPRi, the effects on gene expression should be systematically compared between the two approaches.

We thank the reviewers for this suggestion, which we have taken by deleting an additional two TE+ enhancers. As described now: “We chose two RLTR13D6 elements that displayed >2-fold change in H3K27ac levels upon CRISPRi (Figure 5—figure supplement 3A, D), but with little or no effect on neighbouring gene expression (Figure 5—figure supplement 3B, E). […] Therefore, whilst CRISPR of individual TEs constitutes a more sensitive approach, our CRISPRi data correlates with genetic excision data and is robust with respect to large effects on gene expression”. We have been careful to word our discussion around the roles of TEs as major and critical regulators of gene expression, and are not excluding their potential implication as minor/redundant modulators.

Please also note that the CRISPRi experiment targeted only RLTR13D6 elements, only one of which (*Tdrd12*-associated) had been genetically excised in the original manuscript.

3) The lack of effects on gene expression is not an exclusive property of transposable elements but of enhancers in general when analyzed individually (this is something that the authors should comment on). As the authors state in their Discussion, one possible explanation is the prevalent enhancer redundancy that characterizes the regulation of many genes. The authors should look for any distinguishing genomic features surrounding the few LTRs that indeed did act as enhancers (ATAC-seq, p300, and promoter HiC data (and others). One might predict that LTR copies with significant impact on gene expression might be found in locations with no other host genome enhancers in the vicinity to serve the same function. In other words, perhaps TEs that have truly become important for gene expression have done so once the natural enhancer(s) for the gene have degraded.The authors should then functionally test whether this could explain the lack of relevance at least for one locus. Briefly, they could select one gene whose expression is predicted to be controlled by at least two enhancers with one of them being an LTR element here studied. Then the authors could delete each of those enhancers either individually or together and evaluate gene expression changes.

We agree with the reviewers that enhancer redundancy is a common feature of enhancer-gene networks in general. We now acknowledge this (Discussion, second paragraph), referring to recent work in mouse limb enhancers (PMID: 29420474). Note, however, that the vast majority of TE+ enhancers do not interact with the same genes as TE- enhancers (Figure 3—figure supplement 1C), suggesting a reduced potential for enhancer redundancy involving TEs.

Regarding the search for (epi)genomic features that can predict the regulatory activity of TE+ enhancers, there are simply too few of these elements in our study to make any observations from such effort meaningful. We therefore would prefer to refrain from making any generalisations in this respect.

We also argue that the demonstration of TE+ enhancer redundancy in one example says little about how extensive this phenomenon might be, and that “extensive genetic work will be necessary to evaluate what proportion of TE+ enhancers act in a redundant fashion with other regulatory elements”. We nonetheless attempted to test this possibility at the *Akap12* gene, where a non-TE region with a strong enhancer signature lies downstream of the TE+ enhancer we previously deleted (see Author response image 1). Unfortunately, we only obtained clones with deletions of the TE- enhancer in cells with a wild-type TE+ enhancer, and did not isolate any double KO clones. Deletion of the TE- enhancer had no effect on *Akap12* expression (see Author response image 1), but we could not ascertain whether it acts redundantly with the TE+ enhancer.
